# Parametric Study to Evaluate the Geometry and Coupling Effect on the Efficiency of a Novel FMM Tool Embedded in Cover Concrete for Corrosion Monitoring

Sima Kadkhodazadeh [1,*], Amine Ihamouten [2], David Souriou [3], Xavier Dérobert [4] and David Guilbert [1]

1   Cerema DTer Ouest, ENDSUM Group, 49136 Les Ponts-de-Cé, France
2   Laboratory LAMES, Department MAST, Université Gustave Eiffel, 44344 Bouguenais, France
3   FI-NDT, 44344 Bouguenais, France
4   Laboratoire GeoEND, Département GERS Université Gustave Eiffel, 44344 Bouguenais, France
*   Correspondence: sima.kadkhodazadeh@cerema.fr or sima.kadkhodazadeh@etu.univ-nantes.fr

**Abstract:** Rebar corrosion represents a major threat to the durability of reinforced concrete structures, primarily in marine environments. Various Non-Destructive Evaluations (NDE) have been developed to detect rebar corrosion; although most of these have delivered successful results, a lack of reliable techniques for proper corrosion prognosis still remains. Under the French Research Agency (ANR) project's "LabCom OHMIGOD" framework, we introduce here a novel embedded tool to evaluate the level of cover concrete contamination from aggressive agents responsible for causing corrosion. This tool is divided into two parts: a reactive part exposed to corrosion, and a permanent part protected against corrosion. Using magnetic materials in both parts entails "Functional Magnetic Materials" (FMM) and generates a Magnetic Observable (MO). Through the evolution of corrosion on the reactive part, its magnetic properties become affected, which in turn modifies the MO. By means of regular monitoring of MO variations, it is possible to evaluate the aggressive agent ingress. Consequently, by using a variety of FMM tools placed at different concrete depths, it is possible to indirectly evaluate the rebar corrosion risk. This paper presents a numerical model of the tool employing Ansys software. The underlying objective is to investigate tool accuracy through its key parameters, namely, geometry, relative distance to the receiver, coupling effect, and border effect from the rebar. Simulation results demonstrate that by choosing an efficient geometry for the reactive part (25 mm × 25 mm × 1 mm) and position for the tool (between 1 and 3 mm), both a sufficient MO variation range and a negligible coupling effect can be obtained when the FMM is more than 5 cm from any ferromagnetic material

**Keywords:** corrosion; non-destructive evaluation; magnetic flux density; reinforced concrete; magnetic observable

## 1. Introduction

Reinforced Concrete (RC) has become one of the most widely used materials in civil engineering structures [1]. However, rebar corrosion constitutes a natural drawback that, in the long term, presents strong durability challenges [2]. The carbonation of concrete alters the medium alkalinity, which in turn favors electrochemical corrosion. Furthermore, the penetration of aggressive agents such as chloride, combined with adequate oxygen and water around the steel rebar, serves to accelerate this process [3]. The chemical reactions of iron and oxygen consume the iron and transform it into complex iron oxides and hydroxides, referred to as corrosion products [4–6]. Given the accumulation of corrosion products on the rebar surface, its volume expands and, ultimately, cracks emerge in the concrete that are actually capable of causing final structural failure [7]. It is therefore essential to develop trustworthy prognostic methods

for corrosion in order to reduce the financial costs related to RC structural failure, repair, and maintenance [8].

Much of the research over recent years has focused on monitoring and detecting RC deterioration through Non-Destructive Evaluation (NDE). These techniques are sensitive to RC properties by virtue of measuring concrete permittivity, electrical potential, resistivity value, and mechanical property variations of RC, in association with the particular method's operating principle. According to [9], NDEs including electrochemical methods (such as open circuit potential (OCP) monitoring [10,11]), elastic wave methods (such as Ultrasonic Pulse Velocity (UPV) [12]), and Electro-Magnetic (EM) methods (such as Ground Penetrating Radar (GPR) [13]) have demonstrated significant advances in the detection of rebar corrosion through evaluation of the corrosion rate or by monitoring the progress of micro-cracks and concrete damage as a consequence of corrosion. However, in most of these methods, the results suffer from huge uncertainty levels due to sensitivity to media conditions (i.e., changes in water content, temperature, or the media moisture level); therefore, it is very difficult to interpret their results, thereby necessitating repeat the inspections and calibration of data, according to [9,14,15]. Furthermore, Structural Health Monitoring (SHM) techniques (such as Ag/AgCl electrodes [16] and Radio Frequency Identification i.e., RFID, devices [17]) have been developed that can provide real-time monitoring of aggressive agent ingress, which leads to corrosion, by using embedded devices in concrete. The most important challenges associated with SHM techniques are providing a power source for the embedded devices, a reliable connection between the embedded device and external monitoring system for remote inspection, and the possibility of concrete destruction in case of failure of each of the embedded devices related to maintenance issues. Thus, these devices need more investigation. In addition, none of the above methods can provide reliable, autonomous, and low-power techniques to estimate rebar corrosion before it emerges.

Considering this problem, the French ANR project "LabCom OHMIGOD" aims to develop a novel SHM device for embedding in cover concrete. This tool is referred to as a "Functional Magnetic Material" (FMM) by virtue of being assigned to materials with dedicated magnetic properties.

The FMM includes a reactive part that is sensitive to aggressive agents and rusts rapidly. When aggressive agents reach the FMM, the reactive part is corroded and the magnetic output signature of the FMM's MO is modified as a function of the corrosion state. Finally, an external receiver can measure the MO variation and thus estimate the availability and depth of aggressive agent ingress.

The present study employs numerical modeling to investigate FMM tool performance. The objective herein is to evaluate this performance through key parameters, namely, tool geometry (reactive-part surface area and thickness). It is thus possible to assess the effective geometry of the reactive part, which, in turn, provides for quick and sufficient variation in the MO. Moreover, by using a multi-FMM tool configuration (with differing relative distances to the external receiver), it becomes possible to evaluate the potential of a lateral coupling effect across the tools according to a parametric study. Lastly, analyzing the potential border effect from rebar on tool performance constitutes another objective of this paper. The results allow us to identify the efficient depth at which the FMM tool may be embedded in concrete to counter the minimum influence from rebar during long-term monitoring.

Section 3 will discuss the numerical model of the tool along with the associated Maxwell's equations as the governing analytical expressions. Next, the parametric study will be based on a numerical model for the purpose of evaluating performance in terms of geometric variation and distance to the external receiver, as explained in Section 4. Section 5.1 will present the configuration of a single embedded tool. The objective at this point is to determine the efficient geometric value of the reactive part, as well as the tool's MO variation range with respect to geometry loss. In Sections 5.2 and 5.3,

the potential coupling effect between tools, as well as the border effect from rebar, will be evaluated. Lastly, the results derived from these modeling efforts will be compared in the conclusion of the parametric study with respect to FMM tool performance.

## 2. Principle of FMM

As illustrated in Figure 1, an FMM tool comprises two parts: a reactive part exposed to aggressive agents and a permanent part benefiting from protection by means of a coating against environmental elements, assumed to be permanently intact. The operating principle of this tool is based on an evaluation of the aggressive agent front in cover concrete through corrosion evolution on the FMM tool's reactive part. Consequently, variations in the tool's magnetic properties during rusting of the reactive part offer the possibility to estimate the contamination level by means of a Magnetic Observable (MO) (i.e., magnetic flux density). The underlying hypothesis is that the reactive-part geometry and magnetic properties are gradually modified due to corrosion, which, in turn, also modifies the MO.

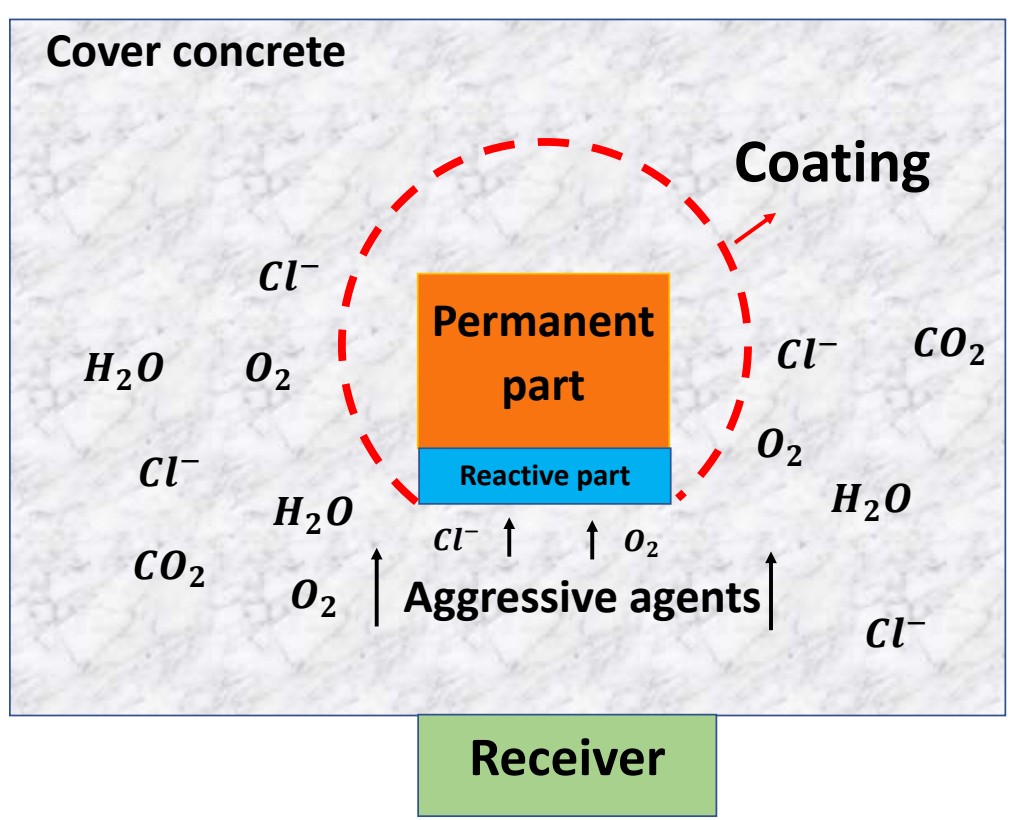

**Figure 1.** Schematic diagram of an FMM tool in cover concrete and the corresponding corrosion scenario: penetration of water, oxygen, and aggressive agents through concrete cracks into the FMM tool, along with protection of the permanent part against exposure to chloride.

Concerning both the tool performance and the correlation between MO variations and the geometric reduction of the reactive part, a feasibility study was initially carried out (see [18]). Thus, experimental results considering the reactive part's geometric variations as an index of corrosion evolution have been validated.

Ultimately, the range of MO variations can be obtained by an external interrogator, which also yields the presence of chlorides causing the reactive part's corrosion, evaluated using non-destructive techniques. As such, an increase in the contamination level in cover concrete could directly lead to RC corrosion; hence, significant MO variation in the tool could be indirectly interpreted as an alert for long-term RC corrosion risk.

The advantages of this notion are twofold. First, the MO remains independent of environmental factors (water content, concrete mixing, or aggregate size) [19], allowing an autonomous, battery-less, and cost-effective tool for long-term and passive monitoring.

Second, by locating the FMM tool at various concrete depths, it becomes possible to estimate the velocity of the aggressive agent front in different cover-concrete zones.

## 3. Numerical FMM Model

Figure 2 presents the numerical model of the FMM tool. Figure 2a shows a schematic diagram of the tool, including the permanent and reactive parts, as well as a coating around the permanent part. The coating is a protective layer made of materials such as epoxy resins [20], which are identified as materials with a magnetic permeability in the range of 1 [21], and are used to prevent any aggressive agents from attacking the permanent part. The coating with non-magnetic properties in Figure 2b illustrates the cubic profile of the FMM tool. Moreover, the total FMM tool size must meet the specific criteria relative to the structural integrity and mechanical behavior of the concrete [22], as stipulated in the patent. By applying a proper mesh size with respect to tool geometry, the simulator divides the total model volume into small sub-regions in order to run a separate Maxwell equation for each region (Figure 2c). The FMM tool output yields the magnetic flux density (**B**) as an MO. Figure 2d–f provide the magnetic flux density propagation with field mapping, iso values, and magnetic counters, respectively. As shown in these figures, a homogeneous magnetic field is generated around the tool. In fact, the permanent part produces a static and uniform magnetic flux line in the environment. However, due to the ferromagnetic materials of the reactive part, the magnetic flux lines are redirected in a dense pattern into the previously magnetized substrate. Thus, as displayed in Figure 2d, at the interface between two parts, the flux lines are more densely pulled into the reactive part.

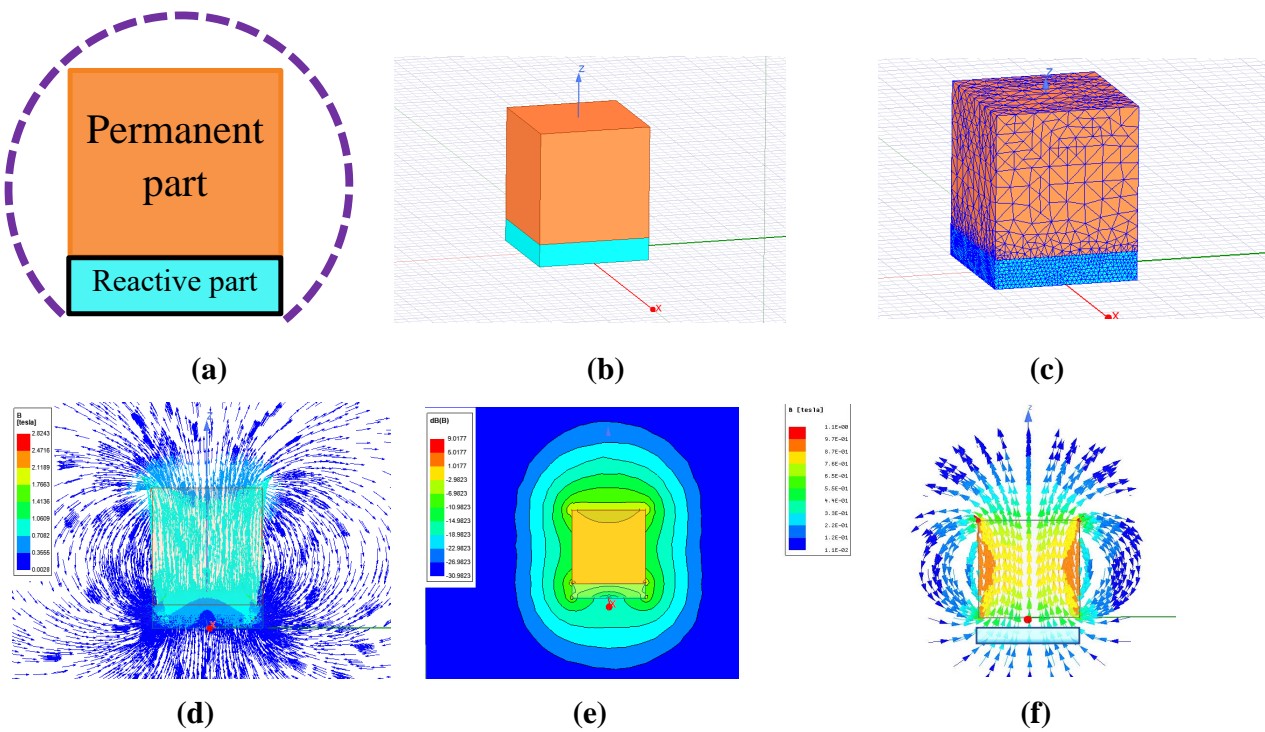

**Figure 2.** Functional Magnetic Materials (FMM): (**a**) general schematic diagram, (**b**) 3D view, (**c**) numerical model with a 1 mm size mesh, (**d**) magnetic flux density propagation, (**e**) magnetic flux density in iso values, and (**f**) magnetic counters.

These flux lines continuously crossing material are tracked by Maxwell's equations, as cited in [23]. The relationship between magnetic flux density **B** and magnetic field strength **H** is governed by:

$$\nabla \cdot \mathbf{B} = 0 \tag{1}$$

$$\nabla \times \mathbf{H} = \mu_0 \mathbf{J} \tag{2}$$

Equation (1) is Gauss's law in magnetism and $\nabla \cdot$ is a divergence operator on $\mathbf{B}$. This relation explains that no magnetic monopole exists. Hence, the magnetic flux line will actually be redirected through magnetized materials to maintain the consistency and continuity between two poles (south/north). In Equation (2), $\nabla \times$ is the curl operator and $\mathbf{J}$ is current density (considered as the source of magnetic field $\mathbf{B}$). Consequently, the relation between $\mathbf{B}$ and $\mathbf{H}$ leads to the following constitutive equation:

$$\mathbf{B} = \mu_0 \mu (\mathbf{H} + \mathbf{M}) \tag{3}$$

$\mathbf{M}$ in Equation (3) expresses the magnetization factor for use on the material interior. Its value equals zero outside of the magnetic (or magnetized) material. Moreover, the $\mu_0 = 4\pi \times 10^{-7}$ F/m term represents the vacuum permeability, while $\mu$ is the relative permeability, which depends on the material's magnetic properties.

By solving Maxwell's equations for the tool model, the simulator calculates the MO from an area around the tool, as illustrated in Figure 2d–f.

All simulations have been run through the 3D module in Ansys software, which relies on the Finite Element Method (FEM) to solve all Maxwell's equations. To solve a set of magneto-static constitutive equations for the FMM tool, the selected geometry must be divided into basic tetrahedral finite elements by means of meshing. The mesh plays an important role in the accuracy of the computed results and thus requires higher resolution in regions containing fields of interest. However, since a student version of Ansys Maxwell was used in this paper, we are limited to 512 K elements.

For the simulations, it is essential to perform an initial parametric set-up for 3D modeling; this step includes choosing the materials and geometry of each component model, along with the relevant material magnetic properties (e.g., permeability) and the meshing size. These parameters determine the accuracy of results and the simulator's computational complexity. Using the Maxwell-3D module, the magnetostatic Maxwell's equation for the passive magnetic flux density, generated by the permanent part and induced through the reactive part, is solved for a finite region (FEM) of space with appropriate boundary conditions among elements and appropriate meshing size, mentioned in Table 1. Finally, when running the simulation, the convergence is achieved successfully which validates the simulation setup. The technical information pertaining to the computer used for these numerical simulations is listed in Table 2.

**Table 1.** General parametric set-up for all objects involved in all three configurations.

| Component Type and Material | Shape | Dimensions (mm) | Relative Permeability |
|---|---|---|---|
| Concrete | Cubic | Half-infinite media | 1 [24] |
| Permanent part of FMM tool | Cylindrical | $\phi$ = 7.5, h = 2 | - |
| Reactive part of FMM tool | Squared | Variable | 50 |
| Rebar (iron) | Cylindrical | Diameter: 20 Length: 200 | 4000 |

**Table 2.** Workstation properties for simulation purposes.

| Processor Type | Intel (R) Xeon(R) CPU ES-1650 |
|---|---|
| CPU frequency | 3.5 GHz |
| Graphics card | NVIDIA NVS 315 |
| Graphics card memory | 4 Go |
| RAM | 32 Go |

The numerical batch models will be examined in the next section for the purpose of assessing tool performance, based on changes to the reactive part's surface area and thickness serving as the geometric parameters, in addition to distance from the external receiver.

## 4. Numerical Model Configurations

Since it has been experimentally validated (in [18]) that FMM tool performance is directly correlated with the reactive-part geometry (e.g., surface and thickness), this paper will solely focus on the reactive part's geometric variations for modeling purposes. The block diagram in Figure 3 provides the numerical study outline in this paper.

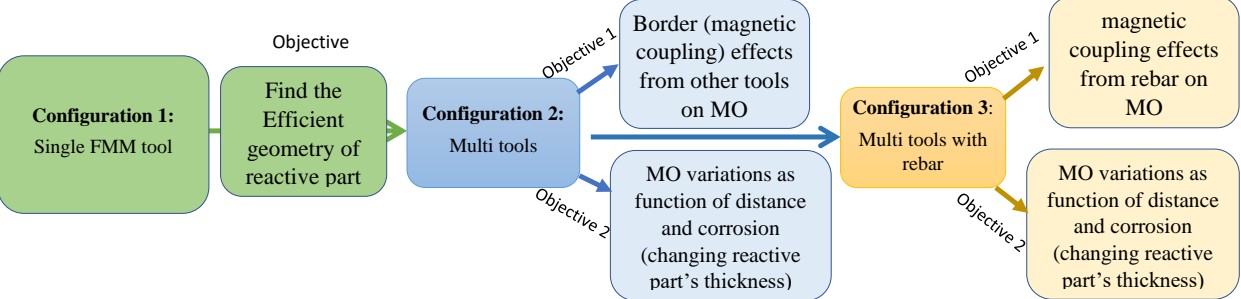

**Figure 3.** Numerical modeling steps and the associated objectives in each configuration.

Three configurations will thus be examined in the following order:

- The first configuration consists of a single FMM tool in concrete, as schematically represented in Figure 4a. A receiver is placed on the bottom surface of the concrete at a distance *r* to the FMM to investigate MO variations through parametric changes, e.g., the reactive part's thickness and width (*h* and *L*), as a function of distance *r*. The main objective here is to find efficient geometry for the reactive part;
- The second configuration consists of three FMM tools arranged at different distances, as diagrammed in Figure 4b. With this configuration, the coupling effect between FMM tools is studied. The influence of changing the reactive part's thickness h (with constant length (*h*) on MO variations is subsequently assessed as well.
- The third configuration is similar to the second, but rebar (ferromagnetic material) is added on top of the concrete. With this configuration, beyond studying the effect of the reactive part's change in thickness on the MO, the border effect from rebar is also explored (Figure 4c).

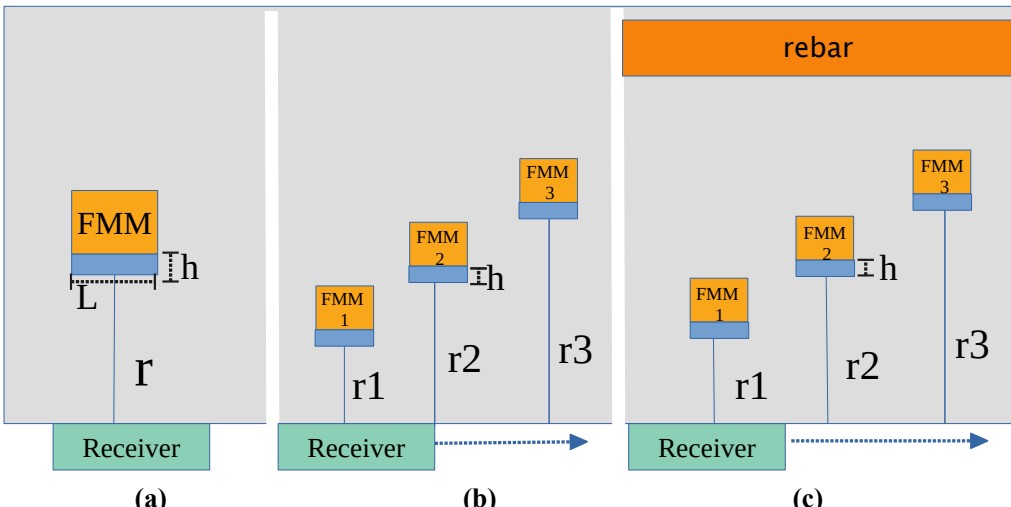

**Figure 4.** Numerical modeling diagrams: (**a**) Configuration 1 with a single FMM sensor, (**b**) Configuration 2 with three FMM tools in concrete, and (**c**) Configuration 3 with three FMM tools in concrete (including rebar).

Table 1 compiles the general parametric set-up of components involved in these three configurations for all of the following simulation steps. Each configuration will be described separately in the next section.

Each configuration will be described separately in the next section.

## 5. Discussion and Results

### 5.1. Configuration 1: Single Tool

Figure 4 displays the schematic diagram of a single FMM tool embedded at a distance *r* in a half-infinite concrete medium, with an external receiver installed on the bottom surface. The objective here is to investigate the efficient geometry of the reactive part associated with an MO variation range produced by a change in the reactive part's dimensions, i.e., thickness and surface area, independently. The notion of effective geometry pertains to the tool dimensions that are capable of providing fast and noteworthy MO variations. The relative permeability of the concrete block, as reported in Table 1, equals 1, which is interpreted as non-magnetic. Furthermore, this value for a square reactive part is set at 50 (value obtained by the material characterization step discussed in Section 3). Due to variable reactive-part geometry during the simulations, the mesh size can be automatically adapted to the dimensions chosen at each step.

In order to visualize the role of reactive-part geometry on the FMM tool output signal, Figure 5 shows the magnetic flux density generated by the FMM tool (featuring arbitrary dimensions), both with and without the reactive part. As illustrated, the magnetic flux line is affected significantly upon removing the reactive part. In fact, more highly permeable materials can alter the spatial distribution of the magnetic field emitted by the external source to an even greater extent [25]. This property ensures that the magnetic flux lines flow along the material and, moreover results, in the diverting of these lines, known as magnetic shielding [26]. Since the FMM tool involves ferromagnetic materials with high permeability, it may be concluded that the reactive part plays a shielding role with respect to the permanent part; Figure 5 displays this magnetic shielding role. In addition, it can be concluded that the evolution of corrosion and the corresponding geometric reduction of the reactive part may lead to the elimination of the initial reactive-part shielding. Eventually, magnetic flux lines will be more likely to propagate into the environment and a magnetic receiver could record the MO rise due to corrosion.

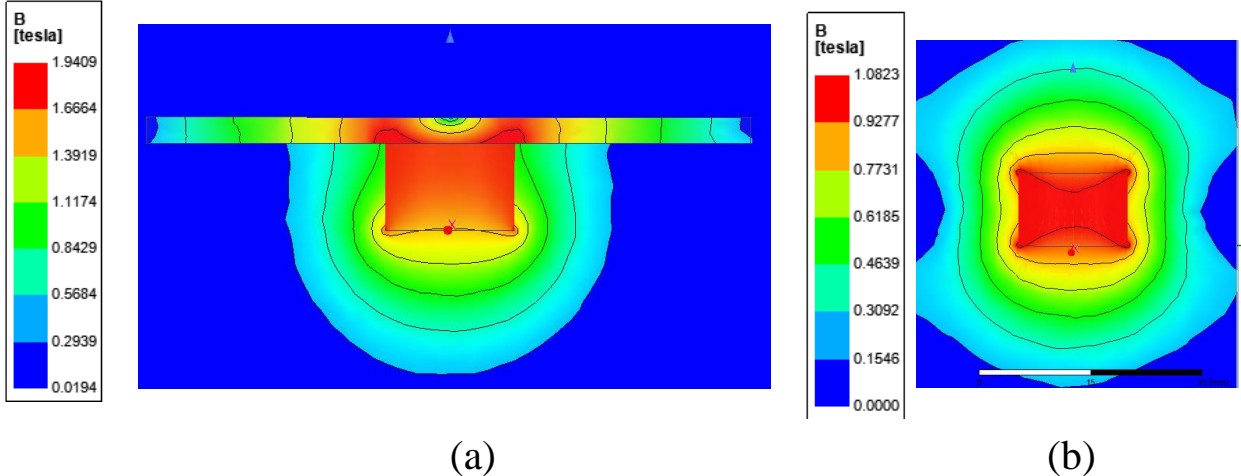

**Figure 5.** (**a**) Magnetic flux density of the FMM tool in the presence of the reactive part as a magnetic shield; (**b**) magnetic flux density of the FMM tool after total removal of the shield (reactive part) by corrosion.

To investigate the MO increment range in greater depth, Figures 6 and 7 illustrate the variation in magnetic flux density **(B)** as the MO, relative to increases in the reactive part's surface area and thickness, respectively, by means of an iso-value-field representation.

For the first simulation, the reactive part's thickness is assumed to have a fixed value equal to 1 mm. Figure 6 shows that by increasing the reactive part length from 8 mm to 18 mm, the MO around the tool starts to decrease. The MO reduction is mainly observed above the reactive part due to the shielding.

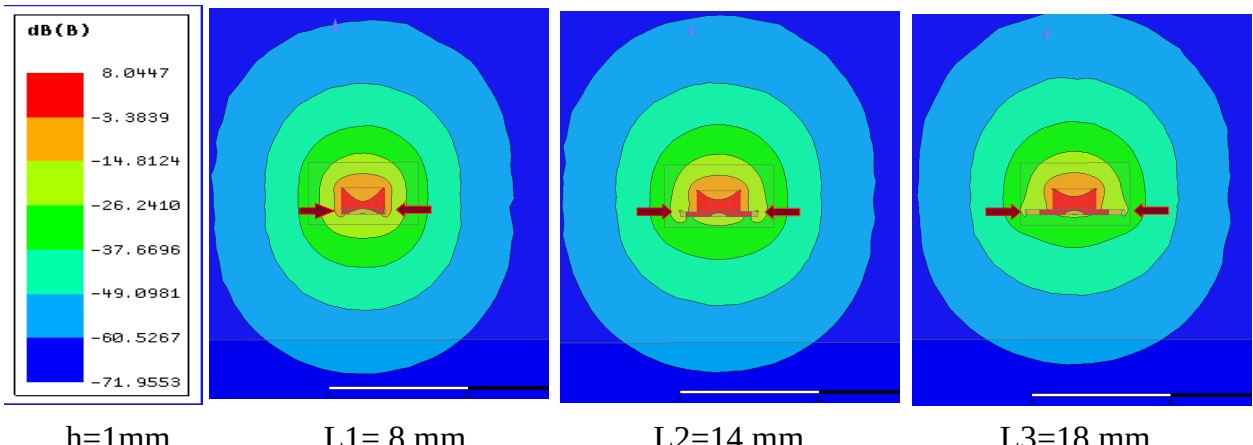

**Figure 6.** MO vs. increasing reactive part length: L1 = 8 mm, L2 = 14 mm, and L3 = 18 mm, with a constant thickness value equal to 1 mm.

In the second simulation, the reactive part's length is assumed to have a constant value of 18 mm. Figure 7 indicates that by increasing the reactive part's thickness from 0.2 mm to 0.8 mm, the MO around the tool is attenuated as well, thus validating that a decrement of the reactive part (either surface area or thickness through any phenomenon such as corrosion) increases the MO propagation. In the following discussion, we will quantify the MO variations for thickness and surface area variations; subsequently, the associated effective geometries will be investigated at each step.

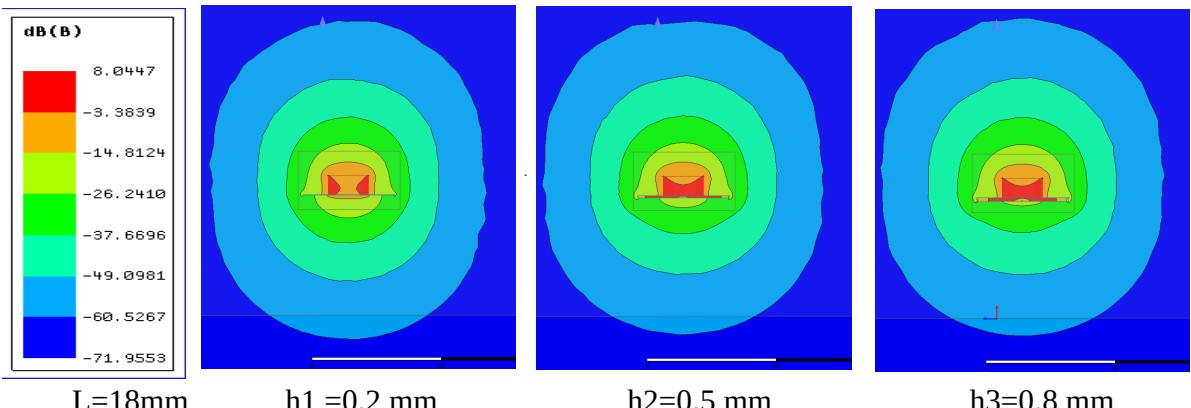

**Figure 7.** MO vs. increasing reactive-part thickness: h1 = 0.2 mm, h2 = 0.5 mm, and h3 = 0.8 mm, with a constant surface-area value equal to 18 mm ×18 mm.

5.1.1. Investigation of the Effective Thickness Value

To quantify the extent of the MO attenuation range and its variations caused by geometric changes as a function of *r*, two additional simulations were performed. First, Figure 8 shows the percentage of relative MO attenuation with thickness increments.

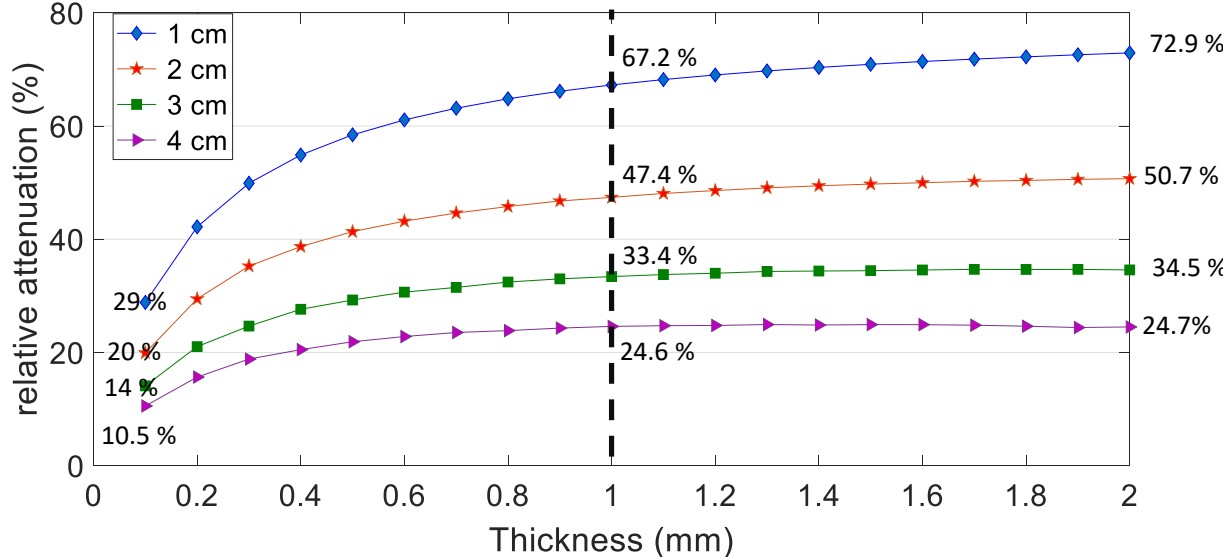

**Figure 8.** Configuration 1: relative attenuation percentage of MO with a constant reactive part value of 25 mm × 25 mm and a variable reactive-part thickness, as derived for four distinct distances between tool and external receiver.

This figure illustrates the MO variation range for the tool with a constant reactive part length *L* equal to 25 mm (surface-area value equal to 25 mm × 25 mm). This value has been arbitrarily chosen. Next, the MO variations can be obtained as a function of thickness from 0.1 mm to 2 mm, with a step size of 0.1 mm. Meanwhile, MO is separately calculated as a function of relative distance, i.e., using the parameter of *r*, which extends from 1 cm to 4 cm with a 1-cm increment.

The percentage of relative MO attenuation is calculated from: *Relative attenuation* $\% = \frac{B_{Permanent\,part} - B_{Reactive\,part}}{B_{Permanent\,part}} \times 100$. Thus, the MO of the permanent part as a reference value is initially obtained as 41.72 mT, 7.79 mT , 2.45 mT, and 1.056 mT, respectively, for relative distances of 1 cm, 2 cm, 3 cm, and 4 cm.

Figure 8 indicates that the relative attenuation graphs have an upward tendency when increasing the reactive-part thickness. Moreover, the range of MO variations is highly

correlated with the relative distance parameter (*r*). For instance, by increasing the reactive part from 0.1 to 2 mm, the MO attenuation is raised from 29% to 72.9%, which is equal to 43.9% (72.9–29%) of the variation range, at the distance of 1 cm. However, this variation range is reduced to 30.7% (50.7–20%), 20.5% (34.5–14%), and 14.2% (24.7–10.5%) for the distances of 2 cm, 3 cm, and 4 cm, respectively.

Moreover, starting from the minimum studied range, all attenuation graphs in Figure 8 reveal a significant growth rate for the thickness increasing to 1 mm (67% out of a total of 72%). Then, the graph slope flattens until 2 mm, thereby supporting the notion that the 1 mm thickness value is briefly effective in obtaining a sufficient range. Similarly, at the distances of 2 cm, 3 cm, and 4 cm, variations of 47.4%, 33.4%, and 24.6% are obtained at a 1 mm thickness, which would qualify as sufficient and rapid MO variations as required. Thus, 70% of MO variations occur by varying the thickness from 0.1 mm to 1 mm. It can be concluded that a thickness of 1 mm is indeed the effective thickness, capable of providing the significant and required MO variations for the FMM tool.

### 5.1.2. Investigation of the Effective Surface Area Value

Featuring a slight modification in the second step, Figure 9 presents the results of MO variations for the FMM tool with constant reactive-part thickness value = 1 mm and variable length, increasing from 15 mm to 30 mm with a 1 mm increment (equivalent to the surface area increasing from 15 mm × 15 mm to 30 mm × 30 mm, considering a square reactive part shape).

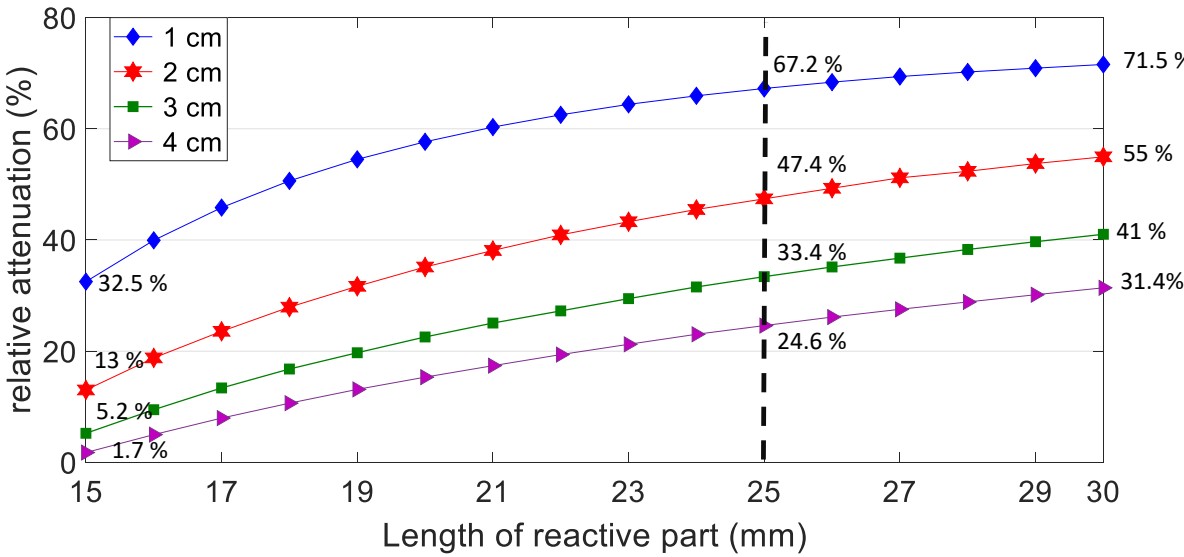

**Figure 9.** Configuration 1: relative MO attenuation percentage vs. the increase in reactive part length, with a constant thickness value = 1 mm, as derived for four distinct distances between tool and external receiver.

The graphs are presented as a function of relative distance. Similar to thickness variations, the direct relationship between the reactive-part surface area and the relative MO attenuation can be obtained. For instance, at the minimum reactive part length = 15 mm, the relative MO attenuation equals 39%, 42%, 35.8%, and 29.7% for distances of 1 cm through 4 cm, respectively. By increasing the reactive part's surface area to 30 mm (via a greater reduction in MO due to stronger shielding), a higher MO would be obtained, i.e., 71.5%, 55%, 41%, and 31.4% at the unit centimeter distances ranging from 1 to 4 cm. Similar to the previous discussion, the variation rate of the graphs in Figure 9 tends to slow after a certain threshold, i.e., at L = 25 mm. For instance, at the 1 cm distance, attenuation rises from 32.5% to 67.2% as caused by increasing the surface area from 15 mm to 25 mm. It may be concluded that the significant and effective inspection range (for MO range monitoring) is manifested by changing the surface area from 15 mm to 25 mm.

Therefore, the surface area of 25 mm × 25 mm is sufficient to obtain fast variations, and is thus an effective surface area value.

In addition, it may be concluded that the effective reactive-part geometry required to obtain considerable MO variations is 25 mm × 25 mm, × 1 mm.

The next section will present a complex FMM tool model consisting of three identical tools with an initial reactive-part geometry, and then compare its output to the results obtained in this section.

### 5.2. Configuration 2: Multiple Tools Used in Concrete

Figure 10 presents the arrangement of three tools embedded at distances of 5 mm, 15 mm, and 25 mm from the external receiver. To study the magnetic coupling effect between tools, the various lateral distances between the tools have been separately studied. Both to save time and maintain the readability of this article, only the results of 70 mm distances between the tools will be presented herein. The FMM tools are horizontally set along the X-axis, as shown in Figure 10. To complete this set-up, the external receiver travels a distance of 200 mm along the X-axis.

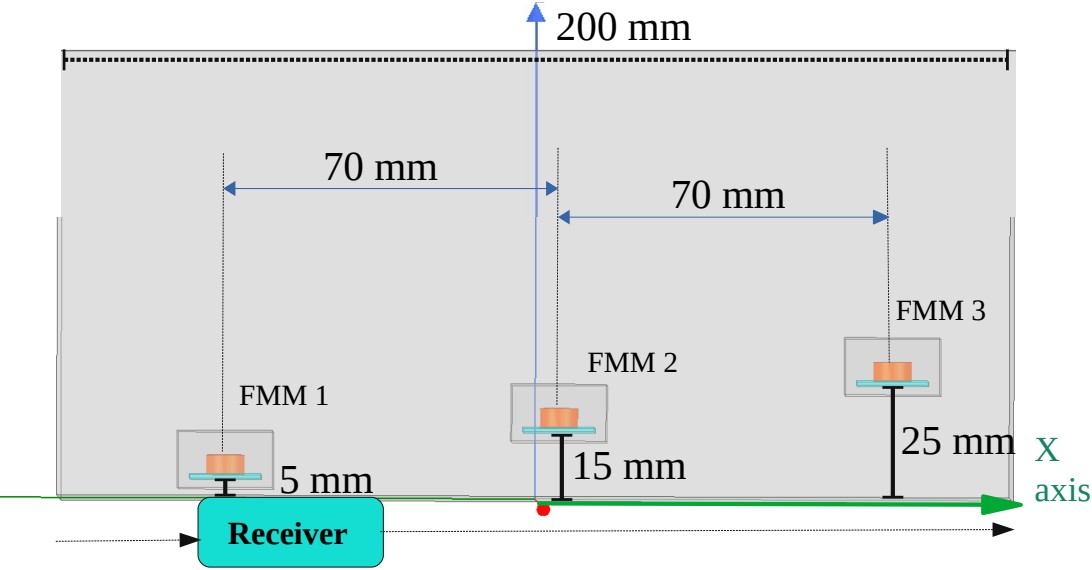

**Figure 10.** Layout of Configuration 2.

The simulation objectives initially call for underscoring the possible coupling effect between tools by using a variety of methods. Secondly, a determination is made of the MO variation range from the multi-tool model with respect to increasing reactive-part thickness. It is important to note that the reactive-part surface area is considered as a constant value equal to 25 mm × 25 mm according to the efficient surface-area value, as displayed in the previous section. In this case, all of the following simulations focus solely on the reactive-part thickness variation. Relative to the first objective, Figure 11 comprises four graphs of MO, as measured by the receiver. The first graph belongs to the overall simulated MO, as simultaneously produced through the three FMM tools. The other three graphs are associated with MO for a single FMM tool independently (i.e., without the presence of the other two tools during simulation). The results in Figure 11 show that the MO values (28, 8, and 3 mT), corresponding to FMM 1-3, are in agreement with those calculated in the previous section. Maintaining a sufficient distance between tools avoids considerable coupling effects on the maximum MO value. By comparing the peaks of graphs associated with FMM 1-3 with the graph for the three tools taken together, the maximum MO values are well adapted for Tools 1 and 2; however, for the furthest tool, i.e., FMM3, a slight difference can be observed at the peak of these graphs, thus implying that coupling effects have a direct correlation with distance *r* as well.

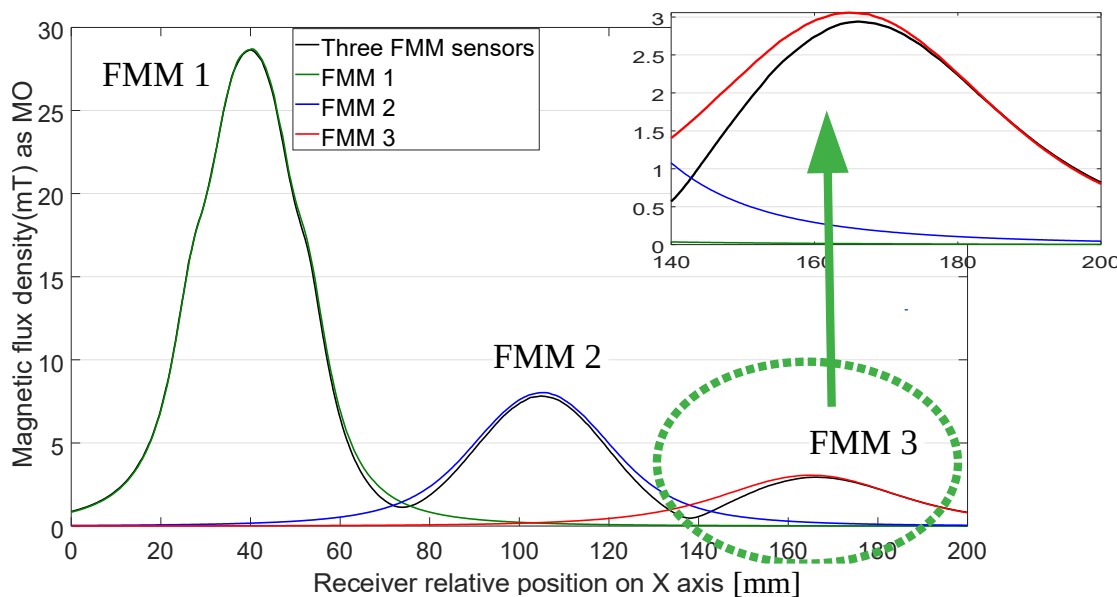

**Figure 11.** Numerical model of Configuration 2: MO for each FMM tool independently vs. MO from all three tools with reactive-part dimensions of 25 mm× 25 mm×1 mm.

Figure 12 compares MO variations along the X-axis with respect to changes in reactive-part thickness, in accordance with the second objective.

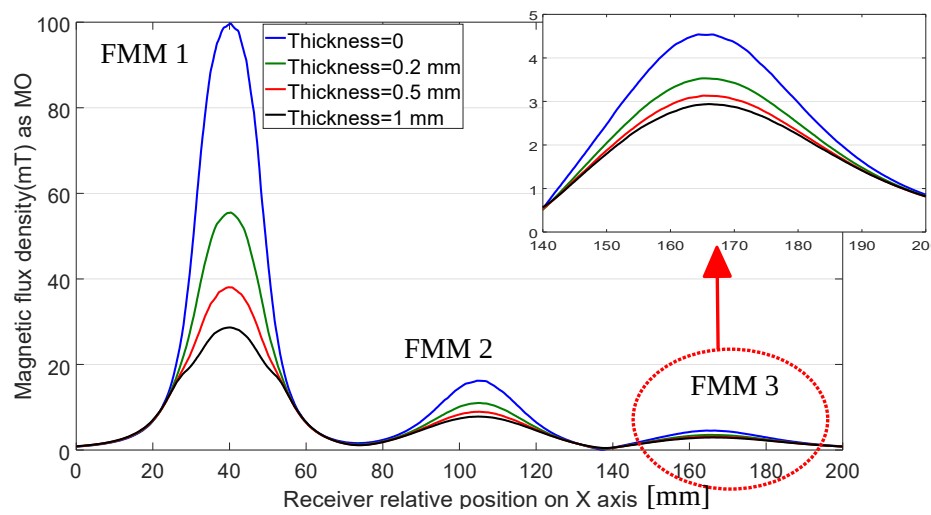

**Figure 12.** Configuration 2: variations in magnetic flux density (MO) relative to a decreasing reactive-part thickness with a constant reactive-part surface area of 25 mm× 25 mm.

By setting the reactive-part thickness at 0 mm, it can be found that the MO value is only produced from the permanent part without any reactive-part contribution; hence, this may be considered as the reference MO value corresponding to each tool. In this case, the reference MO values for FMM 1-3 would equal to 100 mT, 18 mT, and 4.4 mT, according to Figure 12. Simulation results show that peak graph values tend to decrease as the reactive-part thickness increases. These observations are indeed consistent with the results shown in Figure 9, thereby confirming the inverse relationship of the MO and thickness increment. The results displayed in Figure 12 also indicate that the MO variation is more observable for the tool nearest the surface, i.e., FMM1, at a 5 mm distance in the concrete, due to the relatively greater range of MO variation compared to FMM 2 and FMM 3. In this case, the maximum MO drops from 53.5 mT to 28 mT as the thickness increases from 0.2 mm to 1 mm. Consequently, the relative percentage of MO

attenuation rises from 46.2% to 72% with respect to the MO reference value. These results are similar to those of the other tools; however, the total attenuation percentage declines as tool distance increases. By raising the reactive-part thickness to 1 mm, the total relative MO attenuation percentage is therefore lowered to 51% and 34% for FMM2 and FMM3, respectively.

In addition, it can be concluded that when using multi-tools in concrete and maintaining a minimum distance of 70 mm, no significant coupling effect is obtained among FMM devices. These results are indeed extendable to larger concrete samples and, in practice, it can be assumed that by keeping the lateral distances above 70 mm, the tool would be in a safe coupling zone and capable of working independently of other tools. In the next section, Configuration 2 will be extended by adding rebar at a 5 cm distance in the concrete model. This set-up represents a simple cover-concrete model with multiple embedded FMM tools.

### 5.3. Configuration 3: Multiple Tools in Concrete with Rebar

Since the FMM tools are assumed to be embedded in a reinforced concrete structure, the possibility also exists for magnetic coupling between the FMM tool and steel rebar, present as ferromagnetic objects. To investigate the efficient depth in concrete at which the FMM tool can be embedded to receive a negligible border effect from the rebar, the previous design in Configuration 2 is modified by adding rebar at a 5 cm distance. The rebar used in this model has a solid cylindrical shape 20 mm in diameter and 200 mm long, in accordance with Figure 13. As regards the information listed in Table 1, iron with a permeability of 4000 has been assigned as the rebar material and the resulting deduced mesh size is equal to 10 mm. The cover-concrete standards, and specifically the rebar position, have been taken from the French national standards for bridges and reinforced concrete structures in a marine environment [27]. According to this standard, rebar with diameter of 20 mm has been chosen as representative of standard reinforcement in maritime zones presenting significant strain. The main objectives of this numerical model are twofold: (1) to investigate the border effect on MO due to the presence of rebar as a ferromagnetic material, and (2) to assess the MO variation range with respect to the reactive-part-thickness increment.

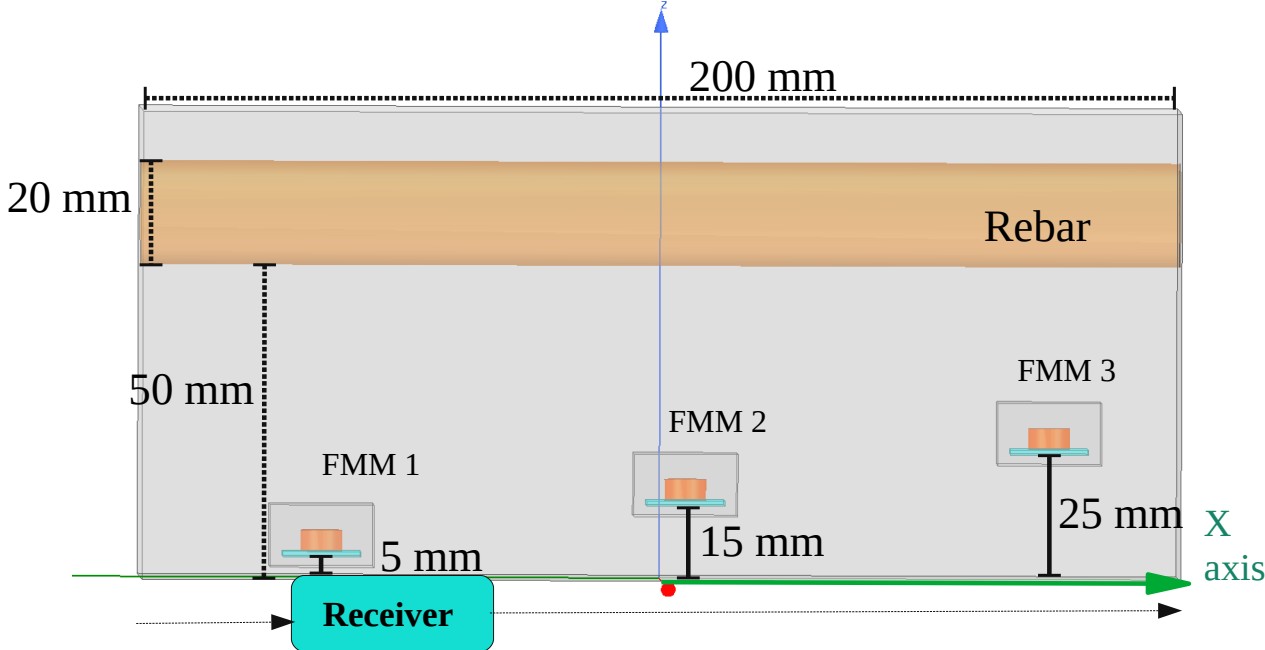

**Figure 13.** Configuration 3: three FMM tools embedded in cover concrete with rebar, featuring variable thickness and a constant reactive-part surface area equal to 25 mm× 25 mm.

Regarding the first objective and similar to Figure 11, Figure 14 compares the overall MO for the three tools, as well as for each individual tool independently, through a moving receiver on the X-axis in the presence of rebar. A comparison drawn between Figures 11 and 14 confirms the previous results of a negligible coupling effect on the maximum MO value for tools located more closely. However, a more detailed observation of the peak around FMM3 shows that by adding rebar, the MO value from a single FMM3 rises compared to the MO from all three tools. These results are confirmed by the negligible border effect of rebar for tools separated at smaller distances. It can be concluded that by introducing a safe lateral distance between tools and then placing them at the center of the concrete rebar mesh, the coupling effect would be avoidable.

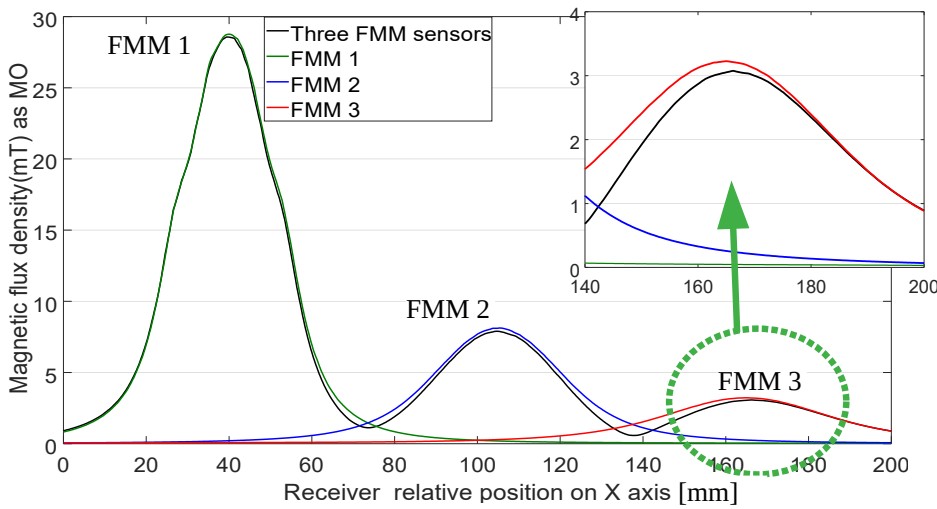

**Figure 14.** Numerical model of Configuration 3: MO for each FMM tool independently vs. MO from all three tools with reactive-part dimensions of 25 mm × 25 mm × 1 mm in the presence of rebar.

The result of MO variations from the three tools relative to reactive-part thickness increasing from 0 to 1 mm in the presence of rebar is illustrated in Figure 15. The inverse relationship between MO variations and increasing reactive-part thickness still agrees with the previous results in Configuration 2. Moreover, the maximum MO value associated with the first tool, FMM1, indicates a declining trend from 54 mT to 29 mT as the reactive-part thickness increases from 0.2 mm to 1 mm. The value for FMM3, the tool closest to the rebar, falls from 3.51 mT to 3.02 mT. For this particular tool, which is slightly more heavily affected by the coupling effect, the percentage of relative attenuation is increased from 22% to 35%; however, this value for Configuration 2 (without rebar) rises from 20% to 34%. Lastly, to include the minimum MO level as obtained by the external receiver during the simulations, it is possible to consider the Signal-to-Noise (S/N) ratio = 20 dB for White Gaussian Noise (WGN) as a reference. The minimum MO is ultimately approx. $+/-0.1$ T, which can be assumed as a detectable threshold in this parametric study.

Comparing the MO from FMM3 in Configurations 2 and 3, the presence of rebar slightly decreases the percentage of relative attenuation. For instance, the MO for FMM 3 with reactive-part thickness = 1 mm is equal to 2.8 mT. This value is raised to 3 mT in the presence of rebar. Therefore, it can be assumed that the presence of rebar results in a small induced magnetic coupling (0.2 mT) for the FMM nearest to the ferromagnetic rebar.

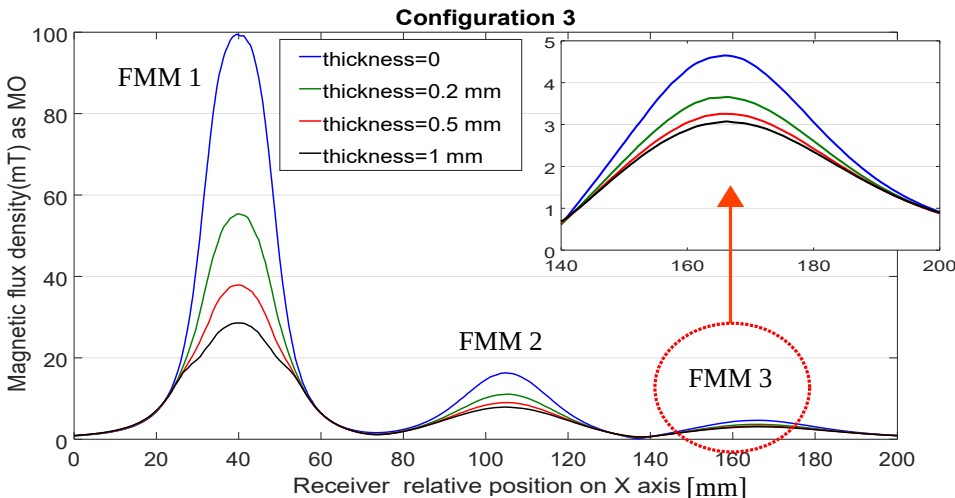

**Figure 15.** Configuration 3: output signal with reactive-part variations in the presence of rebar.

The influence of rebar on the performance of the FMM in Configurations 2 and 3 will be discussed quantitatively in the next section.

### 5.4. Influence of the Rebar Coupling Effect on MO

In order to evaluate the border effect through rebar on the performance of multi-tool configurations, Figure 16 compares the extent of MO variations for three identical tools of dimensions 25 mm × 25 mm × 1 mm, both with and without rebar. The dashed-line graph corresponds to Configuration 3 (with rebar) versus the solid line graph for Configuration 2 (without rebar). These graphs reveal that FMM1 and FMM2 have approximately the same MO values regardless of the presence of rebar, mainly for those tools located at very close distances. However, in the case of FMM3, the MO peak reveals a slightly higher value (2.8 mT vs. 3.0 mT) in the presence of rebar. It can be concluded that by keeping enough distance between magnetic FMM and ferromagnetic rebar, the border-coupling effect on the MO could be avoided. Thus, for the FMM that is 2.5 cm from rebar, the∆ (MO) with and without rebar is 0.2 mT, which corresponds to the rebar coupling.

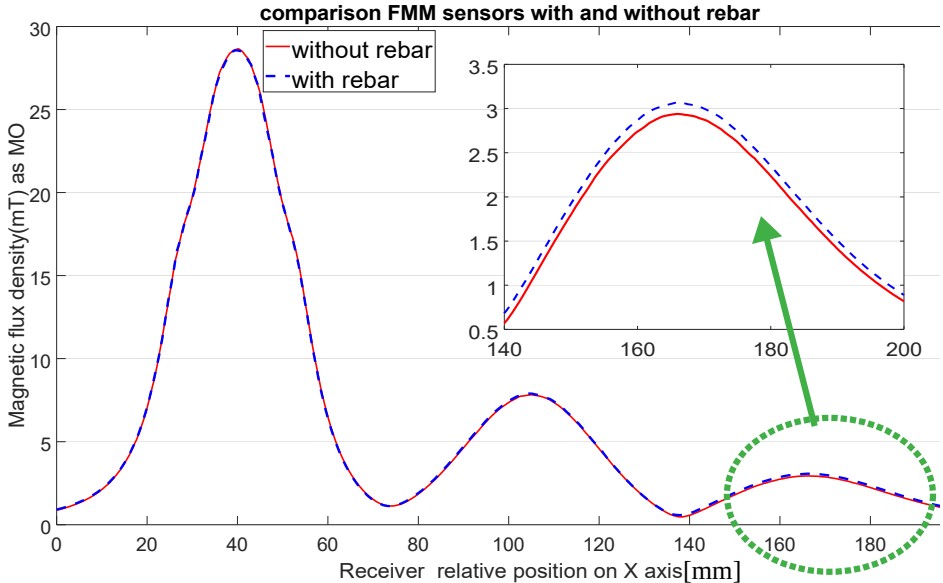

**Figure 16.** Comparison of raw MO values for the multi-tool model when applying an identical tool geometry value: 25 mm × 25 mm × 1 mm for Configurations 2 and 3.

Moreover, Figure 17 compares the percentage of relative MO attenuation for FMM tools with a constant reactive-part surface area equal to 25 mm × 25 mm and a thickness increasing from 0.2 mm to 1 mm. Here, the dashed-line graphs correspond to Configuration 3 versus the solid line for Configuration 2.

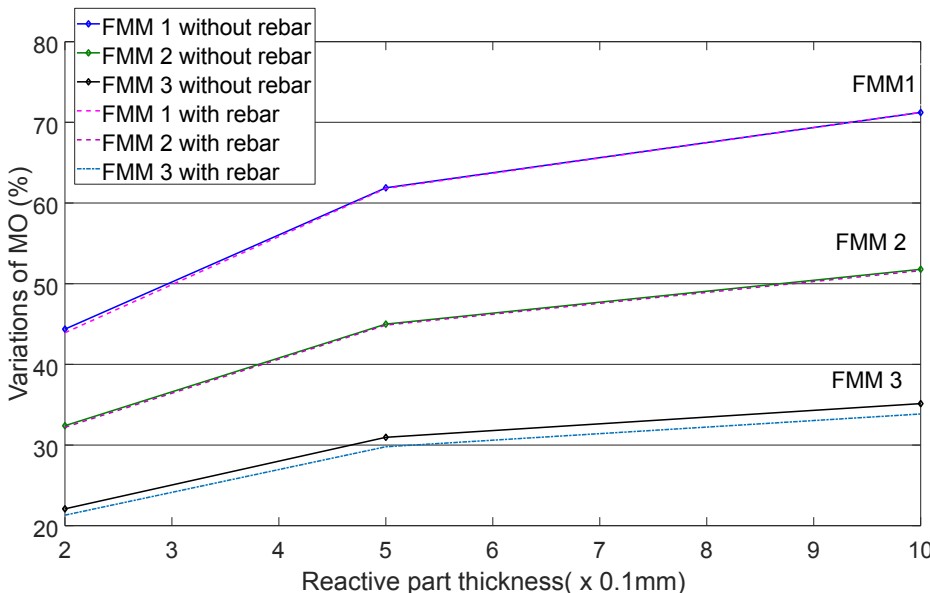

**Figure 17.** Percentage of relative MO attenuation with increasing reactive-part thickness and a constant surface area of 25 mm × 25 mm for Configurations 2 and 3.

The results of the coupling effect on the furthest tool are consistent with the results shown in Figure 9. This demonstrates that the presence of rebar decreases the relative attenuation percentage (less shielding effect) by generating a magnetic coupling between the FMM and rebar. The attenuation is more detectable for FMM 3 by roughly 3%. In addition, the presence of rebar would raise the MO for the FMM device nearest to it by 0.2 mT and, consequently, would change the relative attenuation percentage of the FMM by 3%.

Considering the 3% MO variation, it can be concluded that the minimum coupling effect from rebar could be obtained by keeping the FMM devices at the center of the concrete mesh (minimum lateral distance 7 cm and minimum distance of 3 cm from rebar) to ensure that all of the FMM would remain in a safe area in terms of any type of magnetic coupling.

Further investigations (numerical or experimental) will, however, be essential to determining whether this coupling effect must take into account the Signal-to-Noise (SNR) ratio of the external receiver, depending on the external receiver's physical parameters, minimum environmental noise, etc.

Secondly, the slope of relative MO attenuation increases rapidly as the thickness increases from 0.2 mm to 0.5 mm, but it declines for thicknesses above 0.5 mm in both the FMM1 and FMM2 tools. As mentioned above, the possibility exists to choose an effective reactive-part thickness that yields meaningful MO variations with respect to both external receiver sensitivity and distance *r*. Lastly, the total percentage range of the relative MO attenuation for tools FMM1 and FMM3 with a constant reactive-part surface area lies around 26% (71–45%), 20% (51–31%), and 14% (35–21%) when raising the thickness to 0.2 mm, 0.5 mm, and 1 mm, respectively. These values, which play a significant role in choosing the appropriate external receiver, correspond to measurement sensitivity due to FMM tool geometry, the coupling effect at greater distances, and other environmental elements that must be considered as potential sources of measurement interference.

It is also important to note that for the practical use of this novel tool in concrete during future experiments, a potential level of uncertainty of about +/− 0.5 mm for the distance parameter should also be taken into account.

In conclusion, FMM devices could be embedded at the center of RC meshing at a minimum lateral distance of 7 cm and 5 cm from rebar mesh to ensure that minimum coupling effects would affect MO variations. The simulation results demonstrate the possibility of multi-embedded FMM configurations and MO sensitivity of more than 14% variation for each FMM reactive-part thickness change as an index of corrosion.

## 6. Conclusions

This paper has presented a feasibility study to investigate the performance of a novel SHM tool embedded in cover concrete, used to evaluate the level of aggressive agents in the risky zone of cover concrete. The tool's operating principle is based on the evolution of corrosion in its reactive part, which is capable of modifying MO as a function of geometrical reduction. Using an external receiver, it would be possible to monitor MO variations due to chloride-induced corrosion to estimate rebar corrosion risk, making this an ND technique. In order to investigate the FMM performance and accuracy, a series of parametric studies on the reactive-part geometric variation a corrosion index were carried out through numerical modeling. The objective was to characterize the FMM device through key parameters such as reactive-part efficient geometry, FMM depth in concrete, MO variation range and tendency due to corrosion, and the possible effect of coupling among the FMM and with rebar in the RC model. The simulation results have demonstrated that:

- The reactive part attenuates MO variation due to the magnetic-shielding property of the reactive part;
- Due to the corrosion evolution of the reactive part and its geometrical reduction, the MO increases. The MO increment is highly dependent on the relative distance to the receiver that records the MO. At closer distances, MO variations are more meaningful compared to FMM embedded at greater distances;
- A reactive part with a geometry of 25 mm $\times$ 25 mm $\times$ 1 mm may be considered as an efficient geometric value to generate quick and meaningful MO variations, as required for monitoring the aggressive agent level;
- The coupling effect between tools (when laterally spaced by more than 70 mm) causes a negligible effect (less than 0.5%) on MO variations;
- The presence of rebar for the multi-tool configuration causes a number of MO modifications, specifically for tools placed near the rebar. However, the magnitude of the coupling effect has been calculated to be 0.2 mT (around 3%), which, for purposes of this study, can be considered negligible.
- In practice, to avoid rebar border effects, it is useful to place the FMM tool at the center of the rebar mesh (lateral distance 5 cm) to ensure system performance free of all rebar-induced disturbances.

It should also be mentioned that this paper only presents a feasibility study of the tool's geometric variation as a function of corrosion. Other parameters, such as the role of corrosion-product layers that would be formed on the reactive part during the rusting process and their effect on MO, are proposed for future related studies. Moreover, in the future, additional parameters must be experimentally investigated, e.g., minimum noise level, uncertainty values related to the tool's geometric parameters or distance to interrogator, and the minimum tool sensitivity associated with measurement SNR.

**Author Contributions:** Conceptualization, A.I., X.D., S.K. and D.S.; methodology, S.K., D.S., A.I., X.D. and D.G.; software, S.K., D.G. and A.I.; validation, A.I. and X.D.; formal analysis, S.K. and D.S.; investigation, S.K. and D.S.; resources, S.K., D.S., A.I. and X.D.; data curation, S.K., D.S., A.I. and X.D.; writing—original draft preparation, S.K.; writing—review and editing, S.K., D.S., A.I., X.D. and D.G.; visualization, S.K.; supervision, A.I. and X.D.; project administration, A.I. All authors have read and agreed to the published version of the manuscript.

**Funding:** This research was funded by the French "Agence Nationale de la Recherche" (ANR) through the project 18-LCV2-0002 (https://anr.fr/Projet-ANR-18-LCV2-0002, accessed on 27 August 2022).

**Data Availability Statement:** All data are available upon request.

**Acknowledgments:** The authors are grateful to the French National Research Agency (ANR) for its financial support of the French project "LabCom OHMIGOD".

**Conflicts of Interest:** The authors hereby declare no conflict of interest.

## Abbreviations

| | |
|---|---|
| FMM | Functional Magnetic Materials |
| SHM | Structural Health Monitoring |

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
