# Peer review of "Parametric Study to Evaluate the Geometry and Coupling Effect on the Efficiency of a Novel FMM Tool Embedded in Cover Concrete for Corrosion Monitoring"

_remotesensing, doi:10.3390/rs14215593_

Round 1

Reviewer 1 Report (Previous Reviewer 1)

The authors responded well to the reviewer comments. I recommend this manuscript for publication in Remote Sensing.

Author Response

We thank you very much for your feedback and the time to review our paper.

Reviewer 2 Report (New Reviewer)

Remote sensing

Parametric study to evaluate the geometry and coupling effect on the efficiency of a novel FMM tool embedded in cover concrete for corrosion monitoring

Comments:

This study evaluated the performance of an SHM tool for measuring the level of aggressive agents in the risky concrete zone. Although the research content is sufficient, the research topic is not new and interesting, and also not fits the scope of Remote sensing Journal. Here are some detailed comments for further improvement.

* The text in each figure should be deleted.

* The English editing should be improved, particularly for the cohesion and connections.

* Too much background information was provided, and the main conclusions were not mentioned here.

* The detailed information on the numerical model configurations should be provided and validated.

* The discussion of results should be further conducted. And more quantitative analysis should be added.

* The conclusion should be shortened and more concise.

Author Response

We appreciate you and the reviewers for your precious time in reviewing our paper and providing valuable comments. The authors have carefully considered the comments and tried their best to address every one of them. We hope the manuscript, after careful and extensive revisions, meets your high standards. The authors welcome further constructive comments if any more modifications would be necessary.

We uploaded a pdf file to answer your comments

The authors remain available for all additional remarks and suggestions.

Reviewer 3 Report (New Reviewer)

The paper presents an experimental study to propose a new embedded sensor in concrete for detecting rebar corrosion. The paper is well written and interesting. In our opinion this paper can be published after minor revision. Some remarks are listed below:

-                The abstract need details on the more important results.

-                I method section is needed for more understanding the physical concept.

-            Authors have to reduce the last part of the introduction and added some literature review about embedded sensors in concrete.

-                « Shrinkage is mainly observable under the reactive part » : Are the authors can explain what they means ?

-                From Fig.12 : How authors can explain that FMM1 is more sensitive to FMM2, and 3 ?

-                Whay the diameter 20mm was chosen for testing in the third configuration? is it the more used in concrete?

-                « It should also be mentioned that this paper has merely presented a parametric study of the tool’s geometric variation, yet many environmentally-related parameters have not been 399 discussed ». This sentence is not clear, has to be changed.  

Author Response

We appreciate you and the reviewers for your precious time in reviewing our paper and providing valuable comments. The authors have carefully considered the comments and tried their best to address every one of them. We hope the manuscript, after careful and extensive revisions, meets your high standards. The authors welcome further constructive comments if any more modifications would be necessary.

We uploaded a pdf file, including answer point by point to your suggestions. 

The authors remain available for all additional remarks and suggestions.

Round 2

Reviewer 2 Report (New Reviewer)

Most of comments have been answered and addressed.

This manuscript is a resubmission of an earlier submission. The following is a list of the peer review reports and author responses from that submission.

Round 1

Reviewer 1 Report

The paper is interesting and can be published in the Remote Sensing after considering the following remarks:

1. Line 51 and Fig. 1: more information about the coating is needed.

2. Section 3: More information about FEM modeling is needed, at least the description of constitutive models (concrete, rebar and sensor as well as the interaction between these materials).

3. Section 5.3: From the practical point of view, it would be great to see the verification of the proposed FMM sensor in the real reinforced concrete structure, for example, the RC beam (the real RC beam may be a length of 6 m and height of 0,5 m). The authors presented tests on a relatively small sample and the obtained results may be different for real structure.
Based on the above, the next question arises: how to monitor such a long RC structure using the proposed sensor?

4. The conclusions should be revised and presented the more specific remarks (in bullets) based on the obtained results.

5. Please do not use personal pronouns, some sentences should be rewritten as passive voice.

Reviewer 2 Report

Dear Authors,

I have read your manuscript with great attention and interests.  Overall, the manuscript is poor written. There is not enough scientific contribution in the manuscript, and not good enough to publish.

There are few grammar mistakes and it is not easy to follow the sentences between each session. Unfortunately, the manuscript is not well written to catch the interests and requires additional work and effort. I recommend to publish it after major revisions.

The result and discussion is lacking the scientific explanation especially figure 9 and 10.

The current manuscript is about the corrosion and author did not make any analysis and studies on corrosion product due to different variation parameters.